# Rethinking National Competitiveness for Europe 2050: The Case of EU Countries

**Jurgita Bruneckienė** [1,*] **, Ineta Zykienė** [1] **and Ieva Mičiulienė** [2]

1   School of Economics and Business, Kaunas University of Technology, 44239 Kaunas, Lithuania;
    ineta.zykiene@ktu.lt
2   Faculty of Mathematics and Natural Sciences, Kaunas University of Technology, 44239 Kaunas, Lithuania;
    i.griksaite@gmail.com
*   Correspondence: jurgita.bruneckiene@ktu.lt

**Abstract:** The EU climate policy affects the competitiveness of both the European Union (EU) as a whole and individual member states, leading countries to search for new knowledge to increase their national competitiveness. However, there has been little empirical research about the implementation of green competitive strategies in the common European space from different countries' perspectives. Using the Porter Hypothesis and system theory, this paper explores national competitive strategies that align with climate neutrality in the EU. We used index construction, clusterization, principal components analysis and trajectories change analysis to analyze data from the 24 EU countries from a 10-year period (2012–2021). The main findings reveal three green competitiveness profiles and five green competitiveness progress strategies in the EU. We found that EU countries have different strategies and conditions in terms of their transition towards climate neutrality and competitiveness, which ultimately lead to different rates of progress. Our results provide an initial basis for the development of nation-specific policies to achieve green competitiveness.

**Keywords:** climate policy; national competitiveness; green competitiveness; carbon dioxide; Porter's hypothesis

## 1. Introduction

The EU aims to be competitive and climate-neutral by 2050 [1]. By combining competitiveness and climate neutrality, the EU seeks to simultaneously achieve economic prosperity and environmental sustainability. Such a transition would foster a sustainable and low-carbon economy that not only addresses the urgent challenge of climate change but also ensures long-term economic growth, job creation, and resilience in the face of global environmental and economic transitions. A trade-off between environmental quality and economic growth no longer dominates research or policy narratives and simultaneous targets are identified for growth, sustainability and societal development [2].

Despite the strong commitment that the EU is making, through various communications and recommendations to climate neutrality in the common European space, there are still strong divergences between the results obtained by different countries [3–6]. Giannakitsidou et al. [4] found divergences between Western and Eastern Europe. Škrinjarić [5] found divergences between countries, according to differences in GDP per capita, infrastructure, education, and research and development indicators. Furthermore, Škrinjarić [5] indicates that the most inefficient countries have increased their sustainable development efficiency scores from 2004–2016.

To solve the divergence problem, EU countries are searching for ways to fashion new green growth paths, promote green shifts [7] and allocate large amounts of funding to achieve better results in adapting to climate change and reducing and mitigating its negative effects [6]. Government intervention is seen as a particularly important factor in driving the transition toward climate neutrality in order to prevent engineering and

technological barriers from obstructing the transition [8]. In setting the ambition to solve environmental problems [9], government strategies serve as a catalyst for technological and social innovations, fostering the modernization of industry and infrastructure, as well as behavior changes in society that enhance the competitiveness of businesses and the whole economy globally.

Within this context, the different strategies used by countries to achieve competitive and climate-neutral economies offer an interesting case to investigate. Specifically, EU directives must be applicable at the same scale for all countries, and the local context is essential when fostering transition. This means that all EU members, which may differ in terms of national or regional systems [10], economic and social conditions and policy regimes [11–13], existing economic development paths [14,15], industrial structures [7] or infra-system architectures [16], have to achieve the same goal. However, the transitions from one system to another are often very challenging as existing socio-economic systems, which are often embedded and anchored in specific technologies, infrastructures, institutions and territorial capabilities [14,17–19], which are not relevant to the new system [16]. Economic development and competitiveness advantages are often influenced by past rounds of development, pre-existing economic structures, organisational support structures, institutional set-ups, and natural assets [7]. This means that existing national socio-economic systems do not often automatically lead to new socio-economic systems, and each country may use different strategies in different periods of time to achieve the common EU goal of becoming competitive and climate-neutral. In some cases, existing socio-economic systems may become a trap for green transformation. Klitkou et al. [20,21] defined this situation as a 'lock-in mechanism' and observed that there could be interactions between lock-in mechanisms, such as between learning effects, network externalities, and technological interrelatedness, which reinforce each other, while other interactions could have weakening effects. Kar et al. [22] identified institutional traps, suggesting that low-income countries would not reach the middle-income or high-income level due to the poor quality of institutions. Recognizing that differences in the socio-economic systems may hamper or complicate the timely achievement of common EU goals at the planned scale by all EU members, calls for new knowledge and data about potential strategies to realize the vision of Europe in 2050. Therefore, this study poses the following research question: How can EU countries achieve climate neutrality and competitiveness? Ketels and Porter [23] also recommended a rethinking of the EU policy regarding competitiveness. New knowledge about changes in competitive metrics in EU countries will allow a more accurate reinterpretation of competitive strategies in turbulent scenarios [24].

Much research is being performed on countries' development paths [14,16–18], the progress of geographic sustainability transitions [25], and transformative innovation policies [26]. Various economic shocks, such as the COVID-19 pandemic and the Russian–Ukraine war, has led to a new trend in research: climate change and resilience. For example, Kotseva–Tikova and Dvorak [6] analyzed the green transformation and stabilization of EU member states' economies after the economic slowdown caused by COVID-19, and found that these countries need to take further actions to reduce greenhouse gas emissions. According to Škrinjarić [5] and Hansen, and Coenen [25], there is a need for more systematic research into how place specificity and scale influence transition processes. Theories, strategies, and practices were initially developed in and applied to developed economies, resulting in a lack of knowledge of how they might work in developing and emerging economies. Although there is a wide consensus that place-specificity matters, there is still little generalizable knowledge about how place specificity matters for transitions [25].

We use the Porter Hypothesis (PH) [27,28] and system theory to investigate the competitiveness of countries. The Porter Hypothesis states that the conflict between environmental protection and economic competitiveness is a false dichotomy, and that strict environmental regulation may generate, rather than hamper, competitive advantage. In alignment with this approach, we define climate neutrality or environmental impact as an endogenous factor of competitiveness and define the green competitiveness of a country as economic

(and social) competitiveness, thus leading to an overall reduction in environmental impact (neutrality). The system theory posits that all phenomena can be understood as systems composed of interconnected components that interact both with each other and with their environment.

This paper makes several important contributions. Firstly, the paper contributes to evolutionary economic geography and policy literature, and expands the competitiveness concept by treating environmental performance as an endogenous factor, and clarifies the definition of green competitiveness according to the nature of the relationship between economic competitiveness and climate neutrality. This approach is in alignment with the Europe 2050 vision. Secondly, the paper provides five possible green competitive strategies in the European Union. Thirdly, this paper provides original empirical evidence which may be useful for policymakers creating transformative innovation policies, in order to ensure the progress of coherent geographic sustainability transitions.

The paper is organized as follows. The literature review section presents presents a theoretical analysis of green competitiveness. The research methodology is then introduced. The research results section presents the key findings of the empirical research. The paper ends with a discussion of the results, and the conclusions drawn from them.

## 2. Literature Review

### 2.1. Transformations in the Concept of Competitiveness

The concept of territorial competitiveness is gaining an increasing role in both academic and political-strategic literature, gradually becoming a central theme in governance of the transformation of territories by administrators and decision-makers [29]. As the importance of the concept has grown, the content of the concept has also expanded with various additional factors having been embraced. The evolution of the concept of competitiveness began with a strong relationship with industrialization, infrastructure development, technology, and innovations, and continued with the intensified role of intellectual capital and the creative economy, 'soft' factors, networking and collaboration. The focus of the competitiveness concept has also changed, along with its mission. Following [30] and [31], the core focus shifted from pure economics to a more holistic perspective, involving additional concepts of wellbeing and sustainability. Sgambati [32] found a strong relationship between urban competitiveness and the ability of a city to tackle future challenges, such as climate change, social challenges, resilience, and so on; the ability to adapt to external changes is crucial to developing a competitive advantage, as it allows threats to be transformed into opportunities.

Environmental concerns have also led to significant changes in the concept, attitude, and strategies of competitiveness. In 2015, the World Economic Forum extended its definition of competitiveness to encompass sustainability, defining 'sustainable competitiveness' as the set of institutions, policies and factors that make a nation productive over the longer term, while ensuring social and environmental sustainability [30]. Criticism of traditional economic indicators, such as GDP for offering limited insight into sustainability [31], has led researchers [2,32,33] to apply comprehensive, integrated and holistic approaches to the assessment of competitiveness, and to include sustainability adjustments, including both social and environmental elements, as factors in competitiveness.

Improvements in quality of life and the achievement of sustainable development goals are the dominant objectives of increasing the competitive advantage of a city in all countries, especially in Europe [32]. Two closely related concepts, sustainable competitiveness and green competitiveness, have increasingly aroused scientific interest. Sustainable competitiveness is an umbrella concept that takes a more comprehensive approach to competitiveness, including the economic, social, and environmental aspects of sustainable development, as well as responsible governance. Meanwhile, green competitiveness puts more focus on economic and environmental goals, more precisely reducing carbon emissions or the transition to a low-carbon economy. However, both these concepts focus on long-term competitiveness and well-being. It is important to note that sustainable

competitiveness and green competitiveness are not mutually exclusive, and companies, organizations and institutions often pursue both objectives to varying degrees. Table 1 illustrates the main differences between these two concepts; it should be noted that, due to high interdependency between the two concepts, the criteria can overlap.

**Table 1.** The main differences between the sustainable competitiveness and green competitiveness concepts.

| Criteria | Sustainable Competitiveness | Green Competitiveness |
|---|---|---|
| Pillars | Sustainable competitiveness includes social, economic and environmental development [29] | Balance between economic and environmental development [1] |
| Focus | Balance among different pillars | Finding new areas of competitiveness [34]. Sustainability remains the vital long-term goal, but the Green Economy is described as the pathway to sustainable development [35] |
| Focus | Economic viability, social well-being, and environmental stewardship | Environmental preservation, resource efficiency, clean production and energy, carbon footprint reduction |
| Innovation and Technology | Sustainable innovation and technologies across sectors | Focusing on the development and adoption of green technologies, clean energy solutions, and resource-efficient processes |
| Practices | Total entrepreneurial activity is a driver for sustainable competitiveness [29] | Primarily centered on environmental factors and eco-friendly practices |

The PH states that countries that stick with resource-wasting methods and forgo environmental standards because they are "too expensive" will remain uncompetitive, relegating themselves to poverty [27]. The more stringent policies trigger greater investment in developing new pollution-saving technologies, which induce higher costs for domestic firms in the short run, but the induced innovation will generate economic benefits in the long run by providing domestic firms with a competitive advantage over foreign firms (which will later be constrained by the same regulations) [27]. Several studies [36–39] have proved the PH and found positive effects of innovation on business performance and improved firm resource efficiency on profitability; however, there is a lack of research on whether the first-mover advantage actually leads to improvements in competitiveness in the long run [40].

Policies that promote green growth must be founded on a good understanding of the determinants of green growth and of related trade-offs or synergies [41]. Green competitiveness can be simply defined [33,41] as the ability of a country to maintain or improve its economic performance while simultaneously reducing environmental pressures and contributing to the transition of a low- or zero-carbon economy, or as [1] the ability of a country to provide goods and services that satisfy human needs and improve quality of life while reducing the use of natural resources, the impact on the environment and the emission of greenhouse gases. These definitions combine economic competitiveness with climate neutrality, and limited explanations of the nature of this relationship are provided. The main focus is on ways of achieving green competitiveness, which are highly variegated [7]. Meanwhile, Geyer [42] explained the concept of green business by contributing the 'net green' approach. A business activity is net green if it leads to an overall reduction in environmental impact. By adopting the Geyer [42] idea of 'net green' at the macro-level, we specify the definition of green competitiveness according to the nature of the relationship between economic competitiveness and climate neutrality. We define the green competitiveness of a country as the ability to maintain or improve its economic performance while achieving a general reduction in environmental impact.

### 2.2. The Impact of Place Specificity on Green Competitiveness

Green or sustainable competitiveness is influenced by a multitude of factors, and understanding these factors and their effect on competitiveness can change over time. As knowledge, technology, and societal priorities evolve, the thinking of strategists and policy makers regarding green competitiveness also evolves. New insights, research findings, and emerging trends can shape the strategies and policies implemented to enhance sustainability and competitiveness. Comparative, competitive, and collaborative advantages provide rise to a wide range of policy tools and changes in strategies and plans [43]. It is widely recognized that economic development [18,19] and green competitiveness [44], as well as sustainability transitions [25], are influenced by place specificity. Contextual factors, such as the political environment, innovation, industrial competitiveness, the anticipatory knowledge of local transition managers about transition processes, informal localized institutions, innovation system structures, natural resource endowments, technological and industrial specialization, consumers and local market formation, place-specific norms and values, and many others, have important influences on the geographically uneven landscape of sustainability transition and green competitiveness. Economic development and competitiveness strategies have mainly been analyzed in research related to evolutional economic geography. Martin and Sunley [14] identified strategies based on natural resources, local assets and infrastructure, industrial specialization, technology, economies of agglomeration, regionally specific institutions, social forms and cultural traditions, and interregional linkages; notably, regions may thus employ multiple development strategies. However, it is important to understand that competitiveness goes beyond these indicators or goals [45]. Existing capabilities are the key indicator of future comparative advantages in the green economy [44]. The idea of 'creative destruction', where new ideas drive out the old [46], is central to the type of transformative growth that the green economy discourse espouses [44]. Following Trippl et al. [7] and Fankhauser et al. [44], we treat a country's competitiveness as a reflection of the country's capability to decouple economic growth from environmental degradation and, at the same time, remain a competitive economy.

The past decade has seen a particularly rapid increase in research combining national development strategies and the transition to green or sustainable development, producing heterogeneous results in terms of progress [3–6,47,48] and different ways of achieving it, such as renewal, diversification, importation, and creation [7]. Despotovic et al. [49] stated that post-transition European countries have a lower level of sustainable competitiveness and that heterogeneity also applies to the rest of Europe. Wang and Feng [50] demonstrated that the transition to green development in China is possible but is still progressing at a slow pace.

Global climate policies play an important role in reducing carbon dioxide emissions [47] and many countries have formulated relevant strategies and different policy instruments based on their own or international needs. Zheng et al. [47] also found that different policy instruments have varying levels of performance in reducing carbon dioxide emissions and that while policy effectiveness varies across countries, it is generally higher in developing countries. These heterogeneous findings on the effectiveness of climate policy and competitiveness among different countries demand new knowledge about the possibility of achieving the shared Europe 2050 vision, given the significant differences between specific countries. According to Zheng et al. [47], when a country intends to launch a climate policy which aims to mitigate climate change, it should carefully consider its unique circumstances rather than seeking to replicate the policies adopted by other nations.

Various methods can be used to measure the green (sustainable) competitiveness of a country; however, indices are generally used (for example, the environmental performance index [48] or the green innovation index [44]). These indices, as strategic indicators, reflect the quality of the business environment and influence investors' decisions about entering markets in the indexed country [51]. The idea of the index method, in this case, is to select universal and relevant indicators for all studied countries, calculate each country's performance for each indicator, index them, and rank them according to the index. All

of the important indicators are included in the index function, and the countries can be ranked and compared with each other based on the index results. The index system is necessary to speed up the reform of the system and develop an ecological civilization [52].

## 3. Research Methodology and Data

This section presents the data and methodology used to analyze the impact of EU climate policy on national competitiveness and find strategies to achieve the Europe 2050 vision, which accounts for differences between specific countries.

The research methodology used for impact analysis and the identification of strategies is presented in Figure 1.

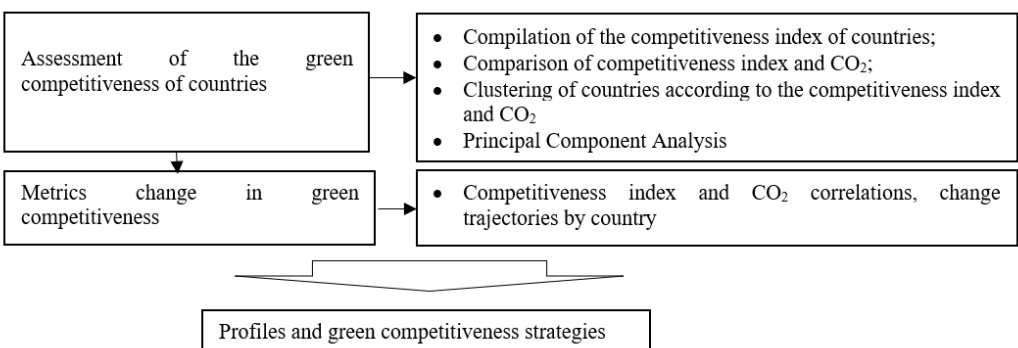

**Figure 1.** Research methodology for impact analysis and identification strategies. Source: designed by the authors.

The research period was 10 years (2012–2021). The units of research were 24 EU countries, having excluded Malta, Luxembourg, and Cyprus due to data limitations. Data was sourced from Eurostat, the World Bank, and Transparency International.

The conceptual framework for the compilation of the competitiveness index was based on the global competitiveness index (GCI) devised by the World Economic Forum (WEF) [8]; however, we focus on the objective (hard) data. We did not include valuations from opinion surveys given to business executives and/or entrepreneurs of countries (soft data), as the methodology of WEF-GCI does. We constructed the competitiveness index from seven groups of factors (pillars) determining competitiveness: (1) institutions, (2) basic and IT infrastructure (combining pillars 2 and 9 of WEF-GCI), (3) macroeconomic environment and market size (combining pillars 3 and 10 of WEF-GCI), (4) health and education (combining pillars 4 and 5 of WEF-GCI), (5) labor market efficiency, (6) innovation, and (7) climate-neutral environment (a new pillar with no equivalent in WEF-GCI). A total of 50 factors were split between these seven pillars. We excluded pillars 6, 8 and 11 from the WEF-GCI due to data availability limitations. The missing values (total 4.93 %) in the data matrix were filled using the Multiple Imputation by Chained Equations (MICE) method. The Pearson correlation coefficient was used to test for correlation between indicators; highly correlated indicators (more than 0.8 or less than −0.8) were removed from the data matrix and a total of 43 indicators remained (a of total 10560 observations). The data matrix and descriptive statistics of the indicators are included in Appendices A and B. The panel data were strongly balanced; to comply with stationarity, we tested the data sample for unit roots. The results of the Levin-Li-Chu test suggested that the variables in the data sample were stationary.

The remaining indicators were normalized using the maximum and minimum value method. The difference between one and the normalized value was used to calculate the indicators with negative influences. No weights were used to calculate the index, based on the practice of researchers [53], who equalize the importance of factors in environmental and sustainability research. Normalized values were summarized (see Formula (1)) by

year, country and competitiveness indices for 10 years, and the 24 countries were ranked accordingly.

$$CI = W_1I + W_2Infr + W_3Ma + W_4HE + W_5LM + W_6Inn + + W_7CNE \qquad (1)$$

where:

- *CI*—competitiveness index;
- *I*—institutions;
- *Infr*—basic and IT infrastructure;
- *Ma*—macroeconomic environment and market size;
- *HE*—health and education;
- *LM*—labour market efficiency;
- *Inn*—innovation;
- *CNE*—climate-neutral environment;
- *Wn*—weight coefficient.

Given the context, the competitiveness of a country was treated as the reflection of the country's capability to decouple economic growth from environmental degradation and, at the same time, remain a competitive economy; 'net green' was evaluated by measurements of $CO_2$ emissions, and was not included as an endogenous factor in the competitiveness index function. In the next steps, the competitiveness indices of the countries were compared with the amount of $CO_2$ generated annually by each country. To identify common trends throughout the EU, the index values and $CO_2$ values were clustered using different methods (k-means, hierarchical method, clustering of continuous data trajectories using the R software (version 4.1.3) library kml3d). After using different internal cluster validation methods (elbow, silhouette, Callinski Harabatz statistics), the optimal number of clusters was determined—5 (see Appendix C).

Principal components analysis (PCA) was used to determine which factors of the 43 selected were the most important for each formed cluster. Before applying the PCA procedure, a Kaisere–Meyere–Olkin (KMO) measurement of sampling adequacy and a Bartlett test for sphericity (p-value) were performed to test for partial correlation and dependence to exclude the potential non-independence of the original data, which can affect the result of a PCA. For PCA to be applicable to a data set, the KMO value should be >0.6. The result of the Bartlett test for sphericity should be significant ($p < 0.05$).

Metric change in green competitiveness was carried out using correlation analysis between the $CO_2$ and the competitiveness indices of the countries in the clusters, and determining the trajectories of the countries' competitiveness indices and $CO_2$ emissions using the Euclidean distance method.

## 4. Research Results

### 4.1. Competitiveness Index and Comparison with $CO_2$ Results

The calculation of the studied countries' competitiveness indices and ranks for 2012–2021 (see Appendix D) revealed that countries develop unevenly in terms of competitiveness. Based on the rankings at the beginning and the end of the research period, the countries can be divided into three groups: (a) countries where competitiveness increased (these countries make up 37.5% of the researched countries, namely Austria, Belgium, Denmark, Estonia, Ireland, Latvia, Poland, Portugal, and Slovenia), decreased (these countries make up 29.2% of the countries analyzed in the study, namely Finland, France, Germany, Italy, Slovakia, Spain, and Sweden) or remained unchanged (33.3% of studied countries, namely Bulgaria, Croatia, the Czech Republic, Greece, Hungary, Lithuania, the Netherlands, and Romania).

The countries' competitiveness indices were compared with the $CO_2$ generated annually by each country (see Figure 2).

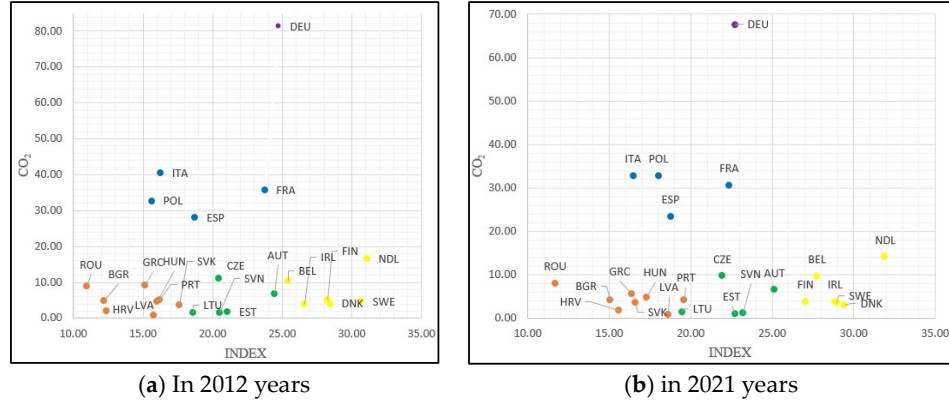

(**a**) In 2012 years          (**b**) in 2021 years

**Figure 2.** The relationship between the countries' competitiveness index and $CO_2$ emissions.

Comparing 2012 and 2021 (see Figure 2), an average reduction of 8% was obtained in the $CO_2$ results. The general situation in the EU is one of positive change: EU countries are changing their economies to be more climate-neutral.

*4.2. Clusterisation*

To detect general trends across the EU and the relationship between competitiveness and $CO_2$, the countries were clustered according to the competitiveness index and $CO_2$ emissions. Five clusters were identified (see Figure 3):

- Red cluster: Bulgaria, Greece, Hungary, Latvia, Romania, Portugal, Slovakia, Croatia;
- Yellow cluster: Belgium, Finland, Denmark, Ireland, Netherlands, Sweden;
- Green cluster: Austria, Czech Republic, Estonia, Lithuania, Slovenia;
- Blue cluster: France, Italy, Poland, Spain;
- Purple cluster: Germany.

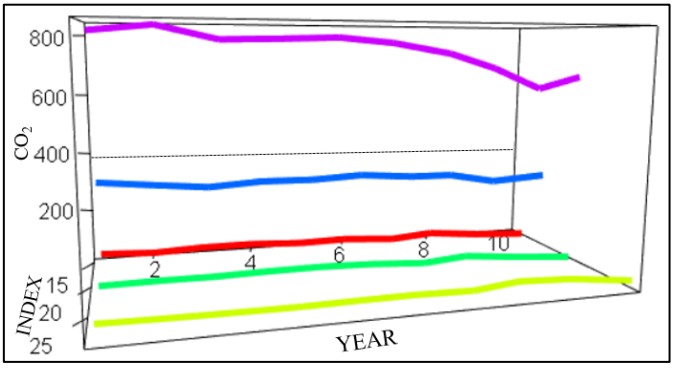
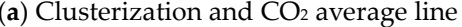
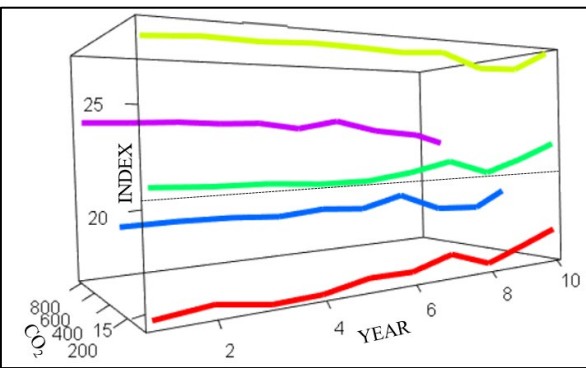

(**a**) Clusterization and $CO_2$ average line

(**b**) Clusterization and competitiveness index average line

**Figure 3.** Results of green competitiveness clustering and matrix. (The figure shows a 10-year period, with 2012 marked as 1 and 2021 as 10).

The competitiveness index was observed to decrease from 2018 (i.e., from the 7th period) and, depending on the cluster, this decline continued until 2019 or 2020 and then began to increase. Only the purple cluster (which contains one country, Germany) could be distinguished from this trend. In this cluster, a decreasing trend of the competitiveness index was observed from 2018, and did not change trajectory in the subsequent period. It should be noted that the $CO_2$ level in Germany decreased quite significantly throughout the analyzed period and only began to increase in the ninth studied year (2020), when the $CO_2$ trends of other groups were more constant or the rate of change was not so drastic.

Following [54] and based on Figure 2, we can identify three profiles of green competitiveness in the matrix of the relationship between the countries' competitiveness index and $CO_2$:

- relatively low competitiveness and relatively low $CO_2$ emissions (competitiveness index and $CO_2$ values are below average);
- relatively high competitiveness and relatively low $CO_2$ emissions (competitiveness index values are above the average and $CO_2$ values are below the average);
- relatively high competitiveness and relatively high $CO_2$ emissions (competitiveness index and $CO_2$ values are above the average);

No countries fit the profile of relatively low competitiveness and relatively high $CO_2$ emissions.

Analysis of the relationship between the countries' competitiveness indices and $CO_2$ emissions showed that no cluster changed its profile between 2012 and 2021.

*4.3. PCA Results*

PCA basically aims to find direct mixes of the factors, which are linear combinations of the original variables (known as principal components) that lead to the directions of maximal discrepancy in the data [55]. This method is used to provide valuable insights into countries' characteristics and identify countries' strategies [56]. In this study, PCA was performed determine the clusters' characteristics (see Appendix E). The results of the PCA show that the clusters of countries are influenced by different factors (see Table 2), which proves that the EU countries have developed differently in terms of competitiveness.

**Table 2.** Green competitiveness factors and profiles.

| Cluster | Cluster Characterization and Competitiveness Factors | Green Competitiveness Profiles |
|---|---|---|
| Red | This cluster is characterised by developing countries with a high reliance on exports (based on indicators in PCA: low export market share per 5 years, low trade of DGP) and a need for economic modernisation (low labour productivity per hour, low GDP per capita). Economic development driven by government stimulation (compliance with EU policy) (no foreign direct investment, low export market share per 5 years, low trade as share of GDP), small and medium-sized enterprises (enterprises per 100,000 people), focusing on core industries—tourism and manufacturing, as well as education. Economically underdeveloped markets, low $CO_2$. Labour-intensive economic activities do not generate $CO_2$ intensively. Government sector influences economic development. This cluster is a mixture of the central and eastern model (Latvia, Romania, Bulgaria, Hungary, Slovakia, Croatia) and the southern/Mediterranean model (Greece, Portugal) [11–13] | Relatively low competitiveness and relatively low $CO_2$ emissions |
| Yellow | This cluster may be characterised by economic development driven by innovation (hight labour productivity per hour, ICT service export, export market shares per 5 years), high-tech exports, productivity oriented towards environmental sustainability, sustainable urban planning (high urban population, low air pollution and high percentage of the population in good or very good health), education and research (high R&D expenditure as share of GDP). Although consumption occurs at a high level, innovations in environmental sustainability and education create the conditions for reducing $CO_2$ emissions from rising consumption and changing the behaviour patterns. This cluster is characterised by the Scandinavian model, which focuses on striking a balance between market relations and state regulation (i.e., welfare states) [11–13]. | Relatively high competitiveness and relatively low $CO_2$ emissions |

**Table 2.** *Cont.*

| Cluster | Cluster Characterization and Competitiveness Factors | Green Competitiveness Profiles |
|---|---|---|
| Green | This cluster is characterised by small open economies, attracting foreign investment (foreign direct investment), diverse economic structure (different kinds of service, manufacturers), export-oriented growth and a focus on digitalisation and innovation adoption (ICT service exports, high-tech export). Economic development driven by government stimulation (compliance with EU policy), productivity and innovation adoption (high average full-time adjusted salary per employee, high labour productivity per hour), export orientation, and education and research (R&D expenditure as share of GDP, health care expenditure as share of GDP). Small markets, relatively low consumption, orientation towards eco-efficiency and exports mean that $CO_2$ emissions are relatively low. This cluster is a mixure of the central and eastern model's 'catching-up' sub-model and the continental model (Austria) [11–13]. | Relatively high competitiveness and relatively low $CO_2$ emissions |
| Blue | This cluster is characterised by big economies and diverse economic structures with sum hubs of high value-added industries. Economic development is promoted by high productivity (high productivity per hour, high GDP per capita, average full time adjusted salary per employee), which is focused on digitalisation (individuals using the internet, R&D expenditure as share of GDP), the introduction of new innovations, and high consumption within countries (high urban population, high household expenditure as share of GDP). Big markets, high consumption, high $CO_2$—but innovations in environmental sustainability and education are offsetting the rate of $CO_2$ increase due to consumption. This cluster is a mixure of the southern/Mediterranean (Italy, Spain); central and eastern (Poland), and continental (France) models [11–13]. | Relatively high competitiveness and relatively low $CO_2$ emissions |
| Purple | Economic development driven by high-productivity manufacturing, oriented towards digitalisation (R&D expenditure as share of GDP) and climate mitigation policies such eco-efficiency and recycling (high packing waste recycling rate, high circular material use rate, high share of renewable energy), education (expenditure on education as share of GDP) and high consumption (household expenditure as share of GDP). It stands out due to its high $CO_2$ emissions due to high production volumes and high levels of consumption. Nevertheless, this cluster is characterised by the fact that production is focused on innovations that are based on digitisation. This cluster is characterised by the Rhenish (German) model [11–13]. | Relatively high competitiveness and relatively high $CO_2$ emissions |

*4.4. Competitiveness Index and $CO_2$ Trajectory Results*

The results for the competitiveness indices and $CO_2$ change trajectories of each cluster are presented in Figure 4.

In addition, correlations between the index and $CO_2$ emissions were calculated, along with the rates of change for each cluster. As the COVID-19 pandemic had a strong impact on the 2020–2021 results, a correlation and change analysis was carried out for the period 2012–2019 to identify strategies for progress in green competitiveness. The correlation coefficient between the competitiveness index and $CO_2$ in the red cluster is (−0.27). On average, the competitiveness indices of the countries in the red cluster grew at a rate of 0.001612, while $CO_2$ emissions grew at a rate of 0.0178. These changes show that the competitiveness index of the countries in the red cluster is increasing at a lower rate than their $CO_2$ emissions. The correlation coefficient for the yellow cluster is 0.66. On average, the competitiveness indices of countries in the yellow cluster increased by 0.00366 units annually, while $CO_2$ emissions rose by 0.01281 annually. The green cluster has a correlation of 0.37 between the competitiveness index and $CO_2$ emissions. On average, green cluster

countries' competitiveness indices varied by 0.00185 and their $CO_2$ emissions by 0.01086. The weak correlation and the rate of change show that as countries' competitiveness increases at a low rate, $CO_2$ decreases at a higher rate. The correlation between the competitiveness indices of the countries in the blue group and the $CO_2$ index is 0.31. The average rate of change of the competitiveness indices of the countries in the blue cluster was 0.00434 and that of their $CO_2$ emissions was 0.01498. The decline in indices is uneven. The correlation coefficient of the variables in the purple cluster is 0.86. The average rate of change of their competitiveness indices was 0.00494 and the average rate of change of their $CO_2$ indices was 0.01942. There was a significant decrease in $CO_2$ emissions.

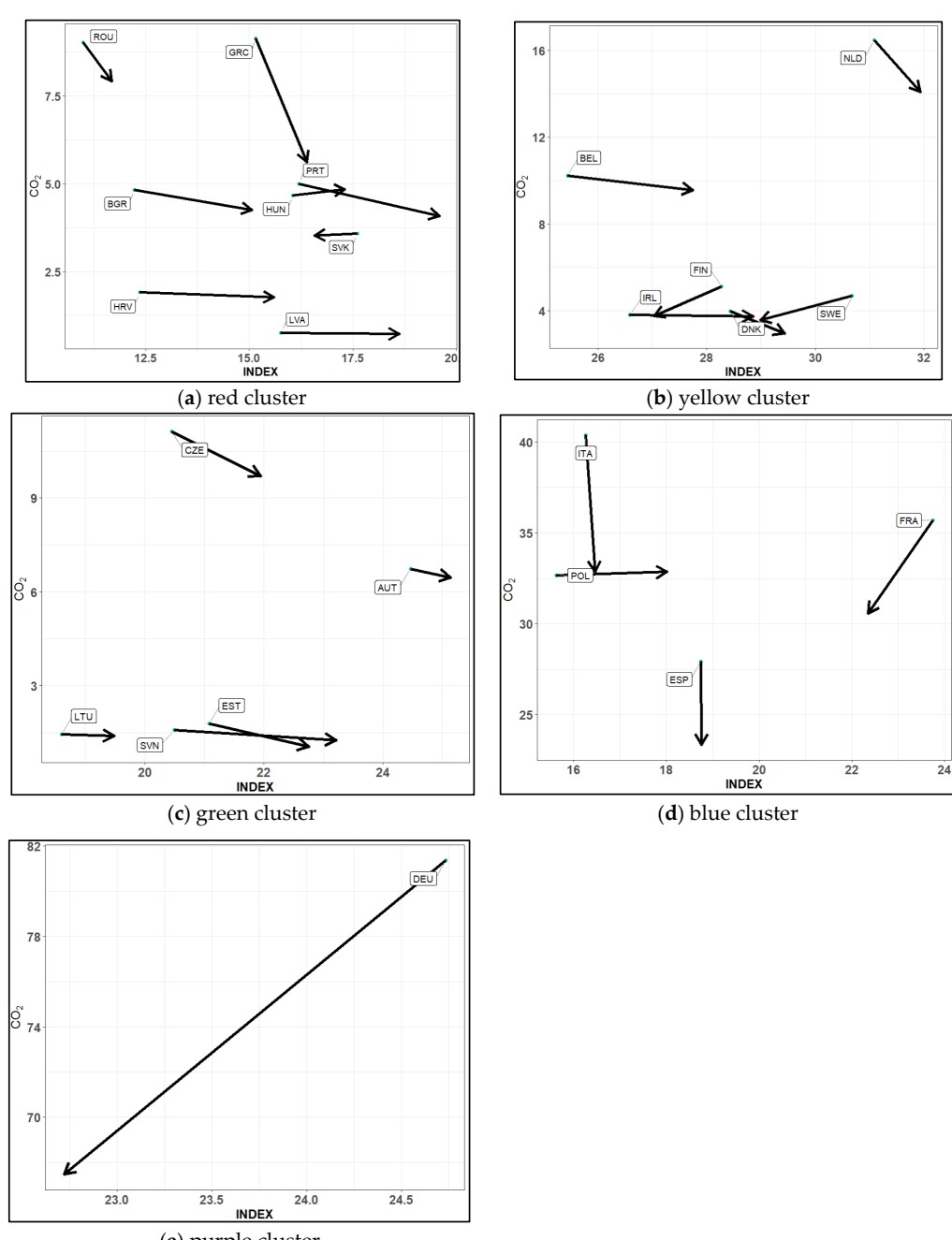

**Figure 4.** Trajectories of EU countries' competitiveness indices and $CO_2$ emissions in 2012–2021. (Euclidean distance).

## 5. Discussion

Based on the results of our empirical research, we have identified five green competitiveness progress strategies which relate to the green competitiveness profiles we determined (see Table 3).

**Table 3.** Matrix of green competitiveness progress strategies.

| | | |
|---|---|---|
| | Profile: relatively high competitiveness and relatively high $CO_2$ emissions<br>Progress strategy (1): Potential growth of green competitiveness<br>Countries (purple cluster): Germany | |
| Profile: relatively low competitiveness and relatively low $CO_2$ emissions<br>Progress strategy (2): lagging increase in green competitiveness<br>Countries (red cluster): Bulgaria, Greece, Hungary, Latvia, Romania, Portugal, Slovakia, Croatia | Profile: relatively high competitiveness and relatively low $CO_2$ emissions<br>Progress strategy (1): Inspiring steady growth in green competitiveness.<br>Countries (yellow cluster): Belgium, Finland, Denmark, Ireland, Netherlands, Sweden<br>Progress strategy (2): Threatening stagnation of green competitiveness.<br>Countries (blue cluster): France, Italy, Poland, Spain<br>Progress strategy (3): Shocking growth in green competitiveness, transitioning to slower growth<br>Countries (green cluster): Austria, Czech Republic, Estonia, Lithuania, Slovenia | Climate neutrality |
| | Competitiveness | |

Countries in the green cluster (with profiles combining relatively high competitiveness with relatively low $CO_2$ emissions) are the most aligned with the Europe 2050 vision and have developed three types of green competitiveness progress strategies: (1) inspiring steady growth in green competitiveness; (2) threatening stagnation of green competitiveness and (3) shocking growth in green competitiveness, transitioning to slower growth. The first strategy represents the greatest progress in economic decoupling. Due to the high level of competitiveness achieved, the rate of increase in countries' competitiveness is relatively low but maintains a strong position, while the rate of $CO_2$ emission reduction is high. The second strategy shows strong progress in economic decoupling; however, it represents a scenario where competitiveness increases at a higher rate than the rate of decrease in $CO_2$ emissions, due to the specific characteristics of countries with big economies. The third strategy shows an initial increase in green competitiveness due to investments in technology (such as clean production and energy) providing potential to increase economic competitiveness and reduce $CO_2$ emissions, which is subsequently followed by declines in the rate of increase in green competitiveness, demonstrating a need for rapid innovations beyond technology (i.e., social innovations, changes in behavior patterns, sustainable urban planning, etc.). Countries in the purple cluster, with a profile of relatively high competitiveness and relatively high $CO_2$ emissions, developed one type of green competitiveness progress strategy: the potential growth of green competitiveness. Due to the high level of competitiveness achieved, the rate of increase in countries' competitiveness is relatively low but maintains a strong position, while the rate of $CO_2$ emission reduction is relatively low due to the large size of the countries' economies. Countries in the red cluster, with a profile of relatively low competitiveness and relatively low $CO_2$ emissions, require significant and long-term investment until the whole economy is competitive and $CO_2$ neutral, and have developed one type of green competitiveness progress strategy: a lagging increase in green competitiveness. In this strategy, the rate of increase in countries' competitiveness is

relatively low, while the rate of $CO_2$ emission reduction is increasing due to investments in technological innovations and labor-intensive activities.

Our research confirms the PH at the macro level and in the case of EU countries. We have clearly identified that the competitiveness of a country can increase in alignment with strict climate policy and EU environmental regulations. These results are consistent with the insights of other researchers who have proved the PH in non-EU counties (macro level). Environmental regulation directly promotes green total factor productivity growth in China [37] and has a positive effect on China's regional development [57]. PH was originally applied at the enterprise level and mainly to large firms [58]. Our research has expanded the PH to the macro level and confirmed that it is relevant at the macro as well as the micro level. Rexhauser and Rammer [38] and Lanoie et al. [39] found that the environmental policy induces economic and environmental innovation that can improve a firm's resource efficiency, but not necessarily its overall efficiency (productivity) to provide positive profitability effects. Positive impacts of environmental regulations have been found on firms and/or industries, which already have a high level of productivity in the most technologically advanced countries [59].

Our research clarifies the PH in terms of green competitiveness profiles and strategies. The identified profiles and green competitiveness strategies provide a good empirical case for narrow PH, which, following Albrizio et al. [59], are less frequently analyzed. A narrow PH points out that flexible regulation provides a stronger incentive for innovation than regular regulation [28]. By identifying different profiles and green competitiveness strategies, we support narrow PH by stating that the specifics of socioeconomic systems (contextual factors and performance level) have an important influence on the effect of environmental regulations and progress in climate neutrality. This is in alignment with Albrizio et al. [59] in the case of OECD countries, and Festa et al. [24] in the case of China.

## 6. Conclusions

The empirical results of the green competitiveness metric changes in the 2012–2021 period in EU countries support the EU political agenda and the current scientific consensus, which states that an economy's gradual transition towards climate neutrality is only visible over time.

The empirical results and the identified green competitiveness profiles support the statement that the competitiveness of countries can be increased in alignment with the strict climate policies and environmental regulations of the EU.

The identification of green competitiveness progress strategies supports the statement that EU countries use different strategies to transition towards climate neutrality, which ultimately leads to different rates of progress. Our results also support the conclusion that countries can shift from one green competitiveness progress strategy to another by adopting climate neutrality development tools and practices. These findings are important for fostering a green competitiveness approach across EU countries.

The identification of different green competitiveness progress strategies also supports the argument that the local context is important for the achievement and success of the EU's common goals. EU countries are subject to different sets of conditions affecting their transition towards climate neutrality, ultimately leading to different rates of progress. This implies a need at the policy level to draw more attention to the need to develop supportive environments (including in terms of infrastructure) and encourage economic subjects to implement practices which will lead towards climate neutrality.

This study has several limitations. The results are specific to the selected sample of countries, which were observed over a defined period and processed using specific types of quantitative analysis. We only analyzed EU countries. Furthermore, we only analyzed a 10-year period, the duration of which is insufficient to identify deeper long-term effects. Averaging and clustering indicators, in certain cases, eliminate the specificity of each country, which requires a more detailed analysis of individual cases. Additionally, we did not include the economic shock produced by the COVID-19 pandemic when



identifying profiles and strategies. Our empirical research also ignores the potential effects of economic and political crises in individual countries, on estimates of their economic and environmental performance.

We identified several future research opportunities. Deeper theoretical and empirical research (including case analysis) on the concept of green competitiveness is needed to help rethink the EU policy on competitiveness. More deeper analysis by the use various mathematical and econometrical methods for identification of trajectories of EU countries' competitiveness indices and $CO_2$ emissions is required. An analysis of how green competitiveness metrics change in response to economic shocks, and what kinds of strategies these changes call for, is also required. Furthermore, incorporating additional countries into the analysis would allow us to compare the green competitiveness profiles and strategies of EU and non-EU countries, and would increase the range of the study field. This would be useful, as the most significant limitation of this work is our restricted study field, which contains only EU economies.

**Author Contributions:** Conceptualization, J.B. and I.Z.; Methodology, J.B.; Validation, J.B. and I.M.; Formal analysis, I.Z.; Investigation, J.B. and I.Z. All authors have read and agreed to the published version of the manuscript.

**Funding:** The project has received funding from European Regional Development Fund (project No. 01.2.2-LMT-K-718-03-0104 under grant agreement with the Research Council of Lithuania (LMTLT).

**Institutional Review Board Statement:** Not applicable.

**Informed Consent Statement:** Not applicable.

**Data Availability Statement:** Not applicable.

**Conflicts of Interest:** The authors declare no conflict of interest.

## Appendix A  Indicators Used in Competitiveness Index Calculation

| No. | Sub-Index | Indicator | Data Source |
|---|---|---|---|
| Institutions | | | |
| 1 | Government Effectiveness | Government Effectiveness Estimate | World Bank |
| 2 | Political Stability and Absence of Violence/Terrorism | Political Stability and Absence of Violence/Terrorism Estimate | World Bank |
| Macroeconomic environment and market size | | | |
| 3 | Gross domestic product | Gross domestic product per capita | Eurostat |
| 4 | Poverty ratio | The at-risk-of-poverty rate is the share of people with an equalized disposable income (after social transfer) below the at-risk-of-poverty threshold, which is set at 60% of the national median equalized disposable income after social transfers. | Eurostat |
| 5 | Foreign direct investment | Foreign direct investment per capita | Eurostat |
| 6 | Level of entrepreneurship | Enterprises per 100,000 capita | Eurostat |
| 7 | International trade as a share of GDP | International trade (% of GDP) | World Bank |
| 8 | Export market shares | Export market shares—5 years % change | Eurostat |
| 9 | Households expenditure of GDP | Households final consumption expenditure (% of GDP) | World Bank |
| 10 | High—technology export | High—technology export (% of goods export) | World Bank |

| No. | Sub-Index | Indicator | Data Source |
|---|---|---|---|
| 11 | ICT service export | ICT service export (% of service exports) | World Bank |
| 12 | Population degree of urbanization | Part of the population living in the city | Eurostat |
| **Basic and IT infrastructure** | | | |
| 13 | Broadband Internet by speed | Broadband Internet speed (Mbps) | Eurostat |
| 14 | Renewable energy consumption | Renewable energy consumption (% of total final energy consumption) | Eurostat |
| 15 | Secure Internet servers | Secure internet servers per 1 million people (TLS/SSL) | World Bank |
| 16 | Internet user | Internet users per1000 people | Eurostat |
| **Innovation** | | | |
| 17 | Total intramural R&D expenditure | Total intramural R&D expenditure (GERD) ratio | Eurostat |
| 18 | Human resources in science and technology | HRST (% from all employed people) | Eurostat |
| 19 | R&D expenditure | R&D expenditure (% of GDP) | World Bank |
| 20 | Patents related to recycling and secondary raw materials | Patents related to recycling and secondary raw materials per 1000 enterprises | Eurostat |
| **Health and education** | | | |
| 21 | Early leavers from education and training | Early leavers from education and training (% of population from 18 to 24 years) | Eurostat |
| 22 | Lifelong learning | Participation in lifelong learning (%of population from 25 to 65 years) | Eurostat |
| 23 | Students enrolled in tertiary education | Students enrolled in tertiary education per 1 million people | Eurostat |
| 24 | Tertiary education | The share of the population with tertiary education | Eurostat |
| 25 | Public expenditure on education | Public expenditure on education (% GDP) | World Bank |
| 26 | People with good or very good perceive health | Share of people with good or very good perceive health | Eurostat |
| 27 | Healthy life expectancy based on self perceived health | Health expectancy in absolute values at birth | Eurostat |
| 28 | Total health care expenditure | Total health care expenditure (% GDP) | World Bank |
| 29 | Self-reported unmet need for medical examination and care | Share of the population self-reported unmet need for medical examination and care | Eurostat |
| 30 | Health impacts of air pollution | Rate of premature deaths due to air pollution | Eurostat |
| 31 | Life expectancy at birth | Mean number of years that a new-born child can expect to live if subjected throughout his life to the current mortality conditions | Eurostat |
| **Labor market efficiency** | | | |
| 32 | Employment rate | Employment rate of the total population | Eurostat |
| 33 | Participation rate | Labor force per working age population | Eurostat |
| 34 | Young people neither in employment nor in education and training | Young people neither in employment nori n education and training (NEET) ratio (% of total population) | Eurostat |



| No. | Sub-Index | Indicator | Data Source |
|---|---|---|---|
| 35 | Labour productivity | Labour productivity per person employed and hour worked | Eurostat |
| 36 | Average full time adjusted salary per employee | Average full time adjusted salary per employee | Eurostat |
| 37 | Fatal accidents at work | Fatal accidents at work per 100,000 workers | Eurostat |
| Climate-neutral environment | | | |
| 38 | Recycling rates for packing waste | Share of recycled packaging waste in all generated packaging waste | Eurostat |
| 39 | Recycling rate of municipal waste | Recycling of municipal waste share to all waste | Eurostat |
| 40 | Exposure to air pollution by particulate matter | The population weighted annual mean concentration of particulate matter at urban background stations in agglomerations | Eurostat, EEA |
| 41 | Air pollution | The emissions intensity of the particulate matter from manufacturing sector in grams per euro of value added | Eurostat |
| 42 | Circular material use rate | Circular material use rate | Eurostat |
| 43 | Grow of forest area | Growth of forest area in 5 years | World Bank |
| 44 | Climate related economic loses | The economic losses from weather and climate—related events, euro per inhabitant | Eurostat, EEA |

**Appendix B  Descriptive Statistics of Data Sample**

| No. | Sub-indicator | Minimum | 1st Quartile | Median | Mean | 3rd Quartile | Maximum |
|---|---|---|---|---|---|---|---|
| 1 | Government Effectiveness | −0.2574 | 0.5889 | 1.0607 | 1.0517 | 1.1527 | 2.2100 |
| 2 | Political Stability and Absence of Violence/Terrorism | −0.2303 | 0.4747 | 0.7485 | 0.6916 | 0.9418 | 1.4008 |
| 3 | Gross domestic product | 5390 | 12752 | 18035 | 24002 | 35480 | 70530 |
| 4 | Poverty ratio | (13.20) 13.20 | (19.10) 19.10 | (22.20) 22.25 | (23.14) 23.15 | (27.85) 27.82 | (38.20) 38.20 |
| 5 | Foreign direct investment | −40.087 | 0.935 | 2.459 | 4.386 | 4.016 | 109.025 |
| 6 | Level of entrepreneurship | (2122) 2122 | (4196) 4228 | (5389) 5381 | (5549) 5517 | (6640) 6611 | (9980) 9980 |
| 7 | International trade as a share of GDP | 54.87 | 81.49 | 104.06 | 116.32 | 150.55 | 252.25 |
| 8 | Export market shares | −32.620 | 9.320 | 1.115 | 1.859 | 9.172 | 78.380 |
| 9 | Households expenditure of GDP | 23.650 | 49.830 | 54.290 | 54.100 | 59.450 | 70.220 |
| 10 | High—technology export | (4.493) 4.493 | (9.398) 9.409 | (12.299) 12.344 | (13.929) 13.994 | (17.678) 17.727 | (32.833) 32.833 |
| 11 | ICT service export | (2.444) 2.444 | (7.485) 7.558 | (9.428) 9.420 | (12.705) 12.688 | (13.949) 13.938 | (58.950) 58.950 |
| 12 | Population degree of urbanization | 53.11 | 61.49 | 70.47 | 71.75 | 80.05 | 98.12 |
| 13 | Broadband Internet by speed | (0.40) 0.40 | (44.27) 42.98 | (63.55) 61.95 | (61.00) 59.54 | (83.53) 82.12 | (98.50) 98.50 |

| No. | Sub-indicator | Minimum | 1st Quartile | Median | Mean | 3rd Quartile | Maximum |
|---|---|---|---|---|---|---|---|
| 14 | Renewable energy consumption | 4.659 | 14.754 | 19.327 | 22.721 | 29.646 | 62.573 |
| 15 | Secure Internet servers | (131.40) 131.40 | (1208.9) 1310.8 | (5333.1) 7168.1 | (18983.7) 23084.2 | (21427.4) 25241 | (277330.6) 277330.6 |
| 16 | Internet user | (45.88) 45.88 | (73.21) 73.26 | (80.69) 80.70 | (79.63) 79.69 | (88.02) 88.11 | (98.87) 98.87 |
| 17 | Total intramural R&D expenditure | (17.80) 17.80 | (28.70) 27.77 | (33.60) 32.75 | (34.32) 33.26 | (39.65) 38.90 | (53.30) 53.30 |
| 18 | Human resources in science and technology | 2.900 | 5.500 | 6.600 | 7.078 | 8.825 | 13.000 |
| 19 | R&D expenditure | (0.3816) 0.3816 | (0.9653) 0.9653 | (1.4108) 1.4108 | (1.7244) 1.7297 | (2.3666) 2.3700 | (3.5272) 3.5272 |
| 20 | Patents related to recycling and secondary raw materials | (0.0000) 0.0000 | (0.1750) 0.1400 | (0.4900) 0.4850 | (0.6703) 0.6678 | (1.0375) 1.0600 | (3.2400) 3.2400 |
| 21 | Early leavers from education and training | 2.20 | 6.40 | 8.45 | 9.27 | 11.80 | 24.70 |
| 22 | Lifelong learning | 0.90 | 5.05 | 8.25 | 10.90 | 14.40 | 34.70 |
| 23 | Students enrolled in tertiary education | (2534) 2534 | (3537) 3585 | (4051) 4037 | (4173) 4168 | (4672) 4663 | (7510) 7510 |
| 24 | Tertiary education | 15.30 | 25.07 | 31.95 | 31.68 | 38.33 | 52.70 |
| 25 | Public expenditure on education | (7.157) 7.157 | (9.616) 9.618 | (10.944) 10.962 | (11.087) 11.096 | (12.300) 12.362 | (18.744) 18.744 |
| 26 | People with good or very good perceive health | (42.80) 42.80 | (59.75) 59.85 | (67.05) 67.15 | (65.57) 65.65 | (72.90) 73.00 | (84.10) 84.10 |
| 27 | Healthy life expectancy based on self perceived health | (60.30) 60.30 | (66.45) 66.50 | (72.10) 72.15 | (70.78) 70.85 | (74.35) 74.62 | (78.70) 78.70 |
| 28 | Total health care expenditure | (4.730) 4.730 | (6.827) 6.817 | (8.670) 8.490 | (8.568) 8.534 | (10.338) 10.340 | (12.820) 12.820 |
| 29 | Self-reported unmet need for medical examination and care | (0.000) 0.000 | (1.125) 1.175 | (2.100) 2.100 | (3.315) 3.307 | (4.300) 4.430 | (16.400) 16.400 |
| 30 | Health impacts of air pollution | (1.00) 1.00 | (34.00) 32.50 | (62.00) 58.50 | (66.50) 64.98 | (95.25) 94.25 | (219.00) 219.00 |
| 31 | Life expectancy at birth | (71.40) 71.40 | (76.95) 76.97 | (80.90) 80.90 | (79.48) 79.50 | (81.70) 81.72 | (84.00) 84.00 |
| 32 | Employment rate | 52.50 | 66.85 | 72.50 | 71.03 | 76.22 | 81.80 |
| 33 | Participation rate | 48.51 | 55.23 | 58.94 | 58.40 | 61.16 | 73.36 |
| 34 | Young people neither in employment nor in education and training | 5.50 | 9.80 | 12.80 | 13.77 | 16.70 | 28.10 |
| 35 | Labour productivity | 42.90 | 64.90 | 78.95 | 91.91 | 119.83 | 209.80 |
| 36 | Average full time adjusted salary per employee | (5266) 5266 | (13102) 13102 | (21620) 21620 | (27415) 27415 | (42125) 42125 | (72247) 72247 |
| 37 | Fatal accidents at work | (0.270) 0.270 | (1.425) 1.387 | (2.115) 2.095 | (2.220) 2.177 | (2.810) 2.763 | (5.780) 5.780 |
| 38 | Recycling rates for packing waste | (36.10) 36.10 | (59.23) 58.67 | (65.55) 64.95 | (64.26) 63.44 | (69.53) 69.33 | (85.30) 85.30 |

| No. | Sub-indicator | Minimum | 1st Quartile | Median | Mean | 3rd Quartile | Maximum |
|---|---|---|---|---|---|---|---|
| 39 | Recycling rate of municipal waste | (10.30) 10.30 | (28.10) 28.48 | (36.30) 36.20 | (37.34) 37.51 | (48.10) 48.12 | (68.30) 68.30 |
| 40 | Exposure to air pollution by particulate matter | (0.0100) 0.0100 | (0.0500) 0.0500 | (0.0900) 0.0900 | (0.1783) 0.1708 | (0.2400) 0.2300 | (1.0200) 1.0200 |
| 41 | Air pollution | 4.80 | 10.40 | 14.10 | 14.56 | 19.00 | 29.30 |
| 42 | Circular material use rate | 1.300 | 4.300 | 7.400 | 9.067 | 11.600 | 33.800 |
| 43 | Grow of forest area | (−0.9676) −0.9676 | (0.0000) 0.0000 | (0.03512) 0.3512 | (0.06066) 0.06066 | (0.09979) 0.09979 | (0.53171) 0.53171 |
| 44 | Climate related economic loses | 20.60 | 57.40 | 72.30 | 74.34 | 87.53 | 138.80 |

() data before missing data imputation by MICE method.

## Appendix C  Optimal Number of Clusters in kml3d, Callinski Harabatz Statistics

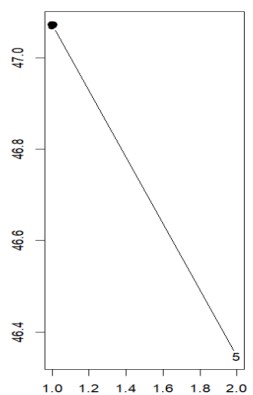

**Figure A1.** Optimal number of clusters in kml3d, Callinski Harabatz statistics.

## Appendix D  Countries' Competitiveness Indices and Ranks

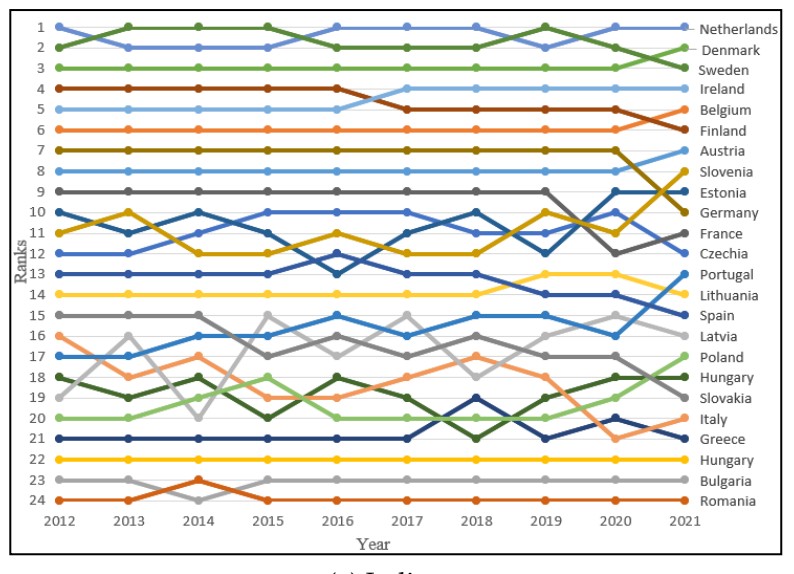

**(a)** Indices

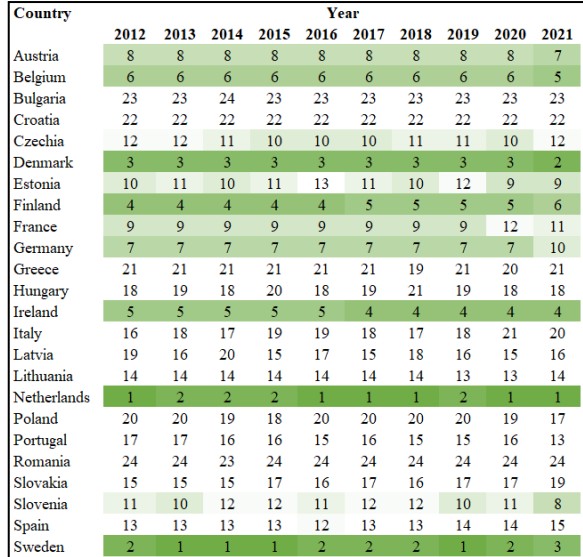

**(b)** Ranks

**Figure A2.** Countries' competitiveness indices and ranks, 2012—2021.

**Appendix E  Decomposition of the Total Inertia on PCA Components (%)**

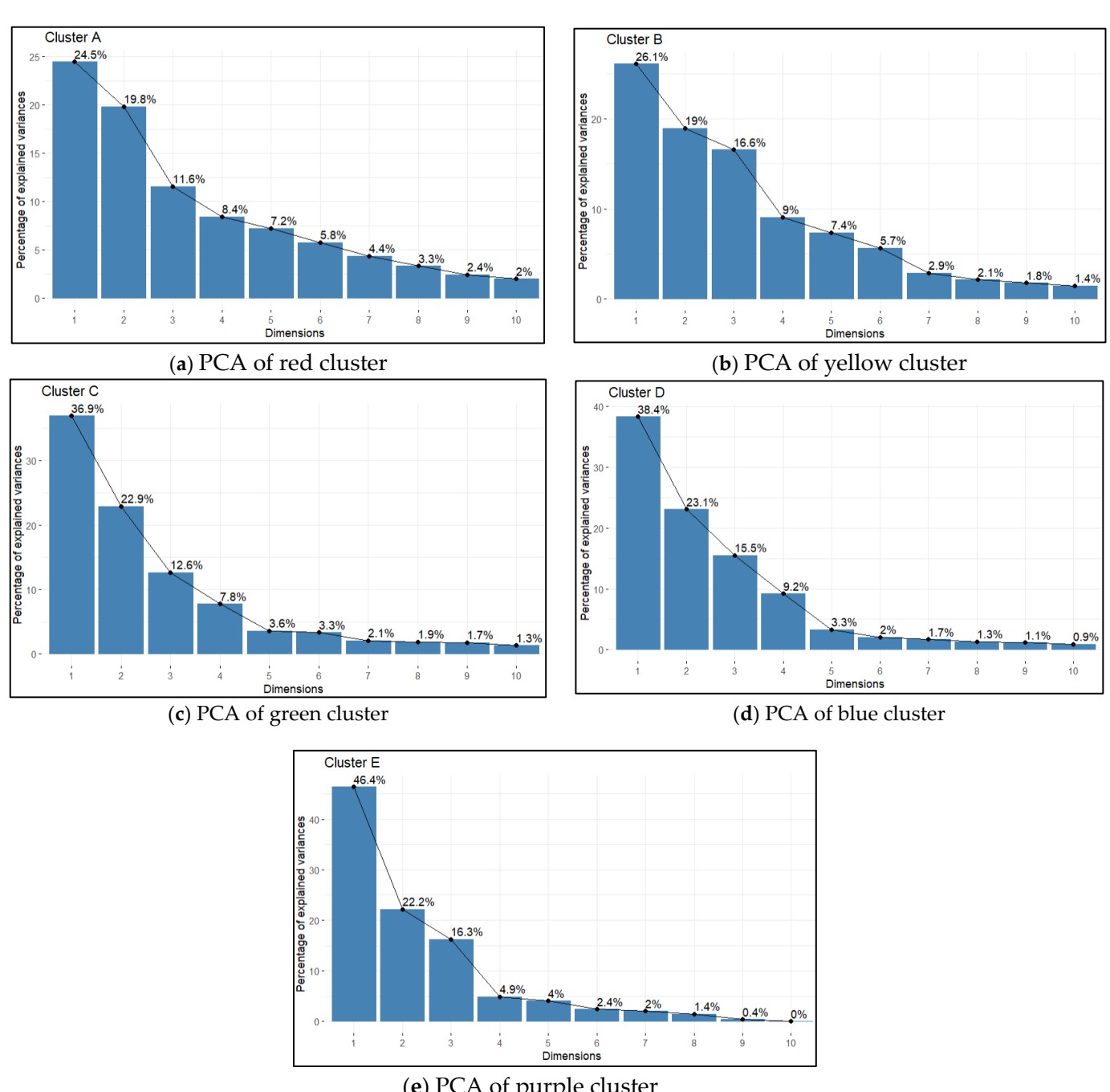

**Figure A3.** Decomposition of the Total Inertia on PCA Components (%).

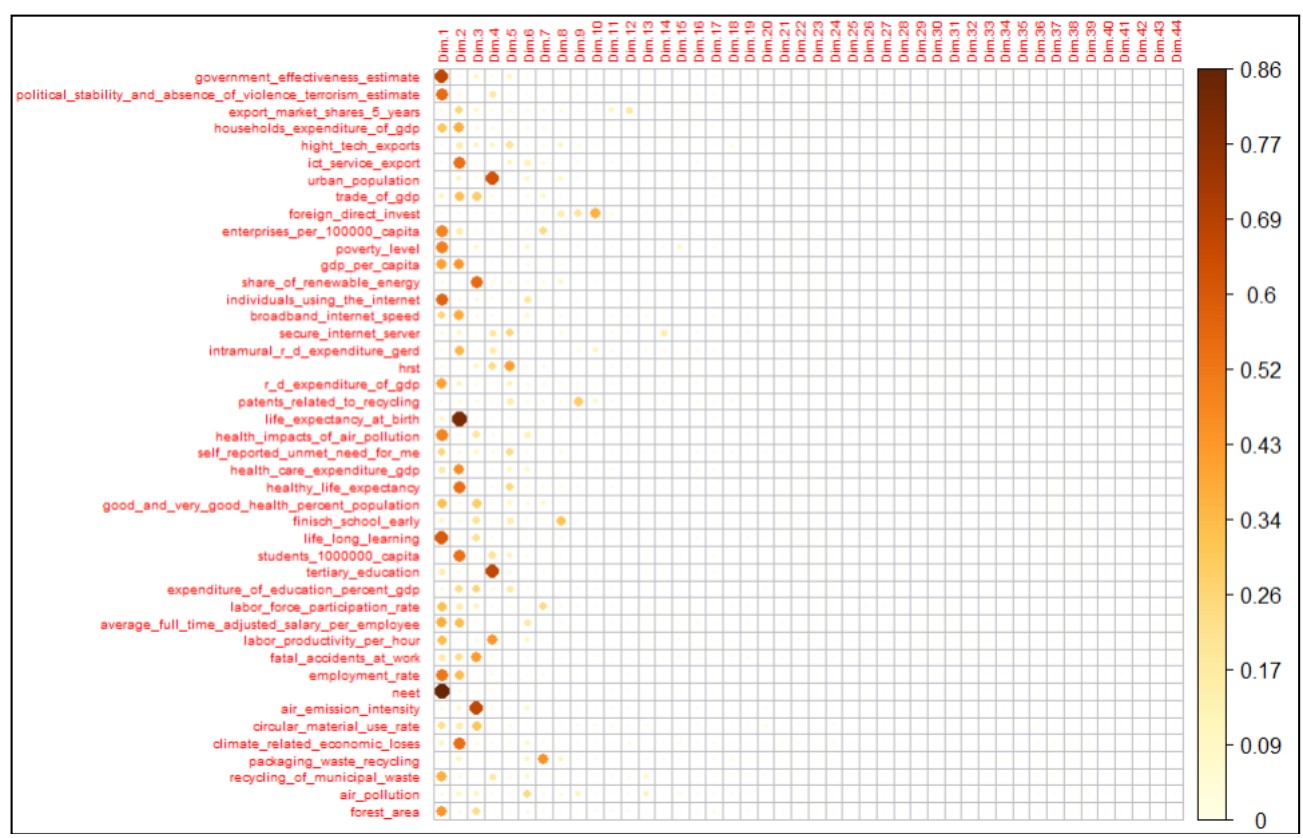

(**a**) PCA of red cluster

**Figure A4.** *Cont.*

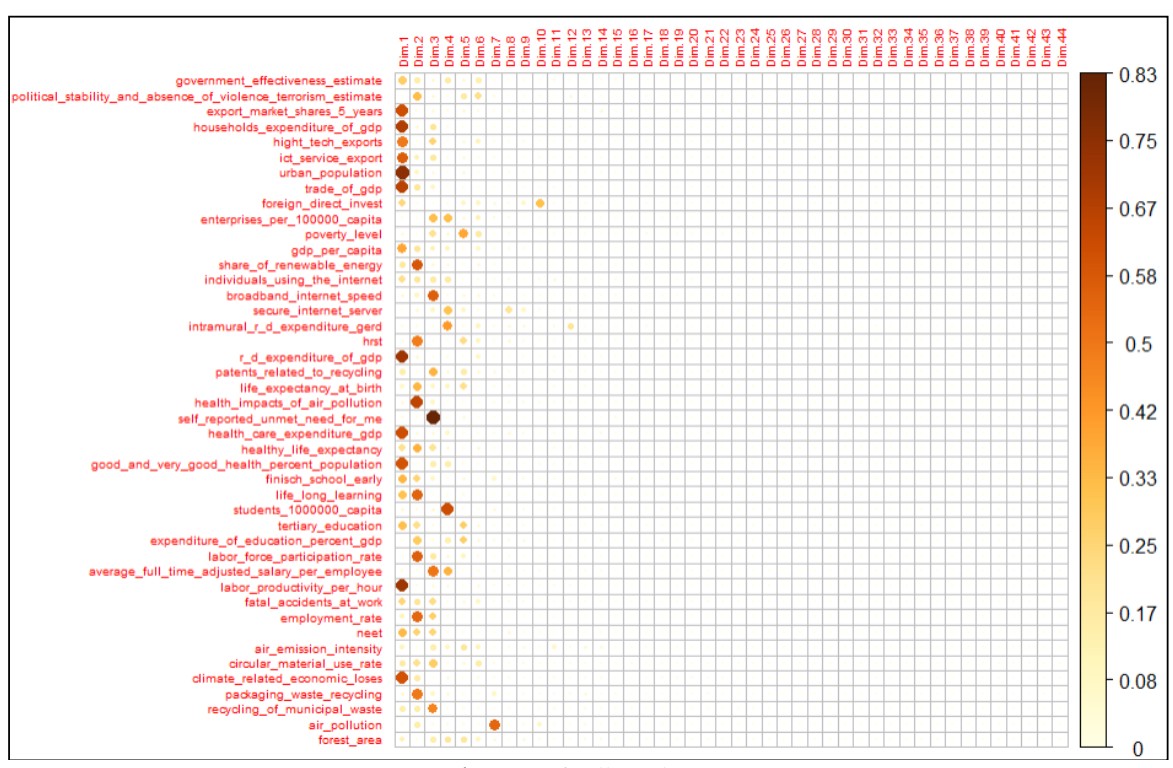

(**b**) PCA of yellow cluster

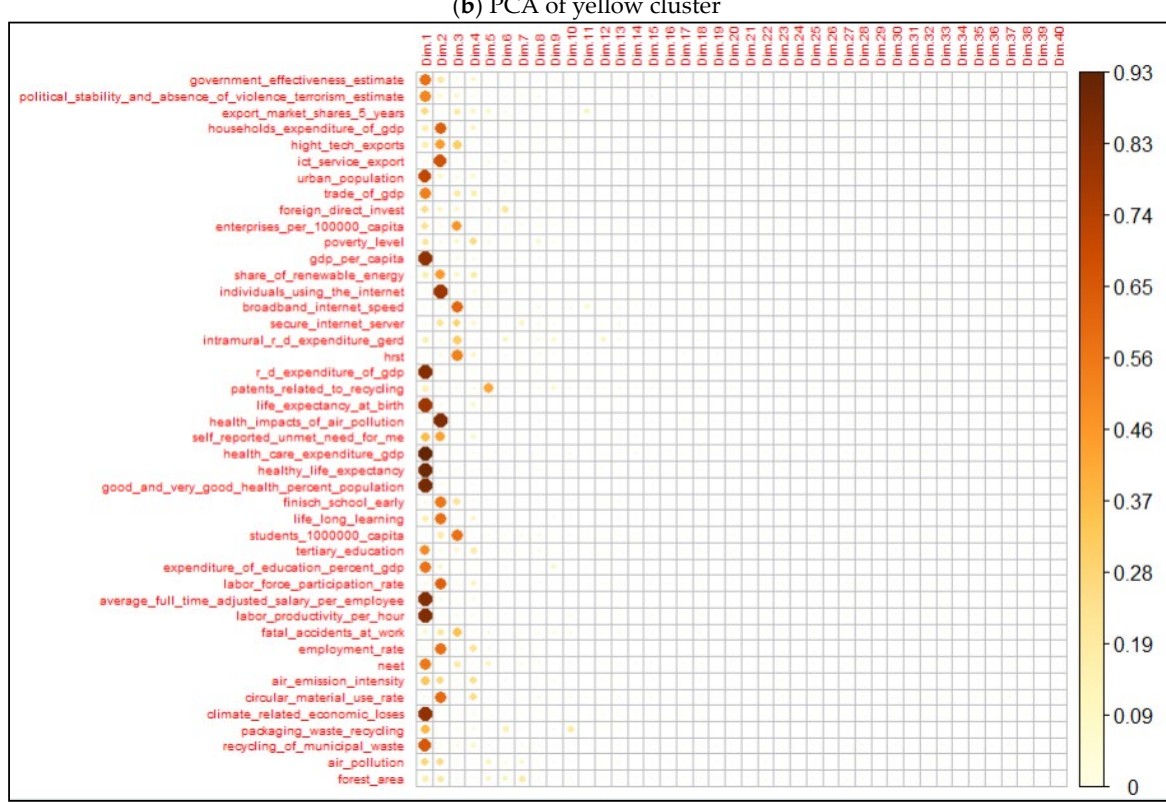

(**c**) PCA of green cluster

**Figure A4.** *Cont.*

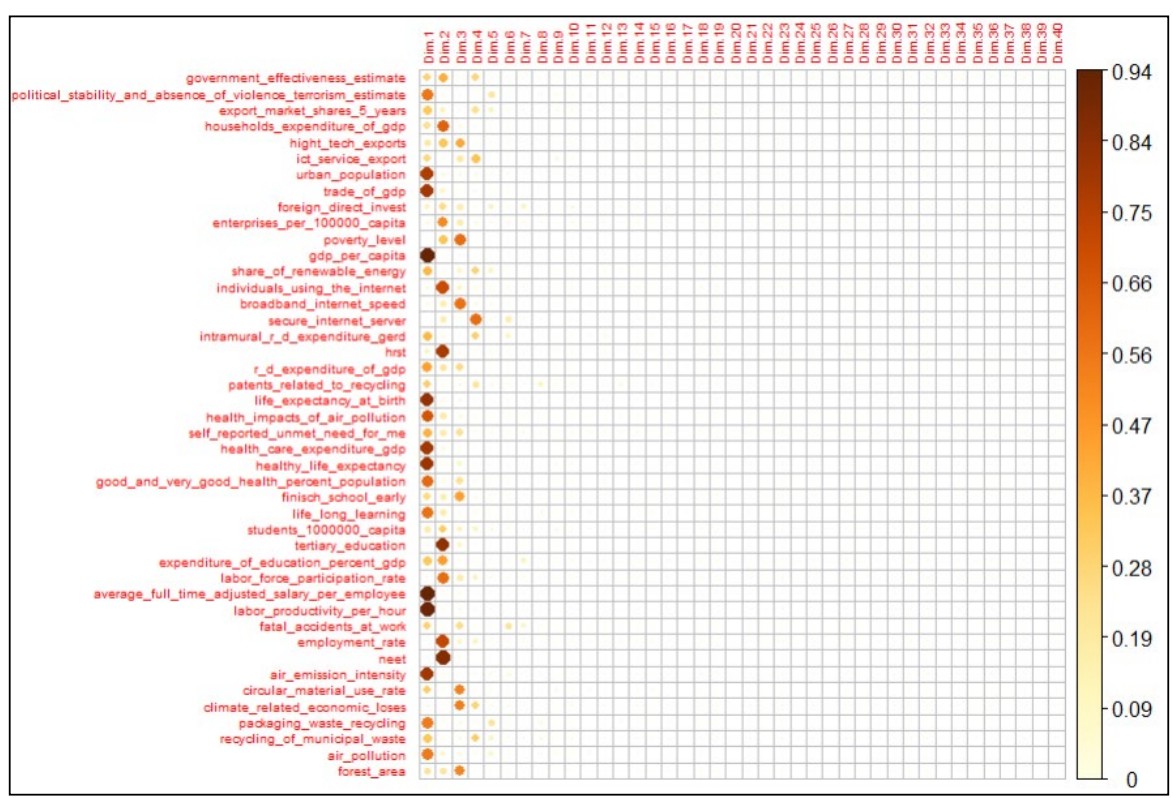

(**d**) PCA of blue cluster

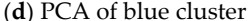

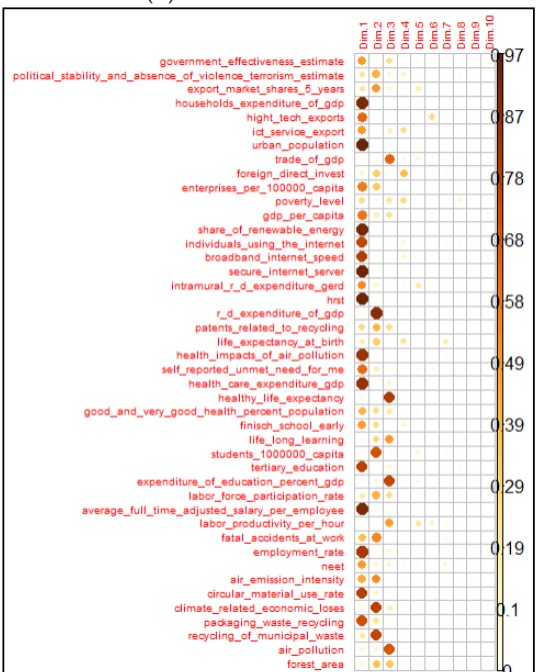

(**e**) PCA of purple cluster

**Figure A4.** PCA of clusters.

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
