# Peer review of "Rethinking National Competitiveness for Europe 2050: The Case of EU Countries"

_sustainability, doi:10.3390/su151310697_

Round 1

Reviewer 1 Report

Some comments (strengths and recommendations) regarding the paper are presented below.

Strenghts:

-        Title of the paper is clear and the topic of the paper is important nowadays;

-        The content of the paper (internal organization) is well structured and has clarity;

-        The paper includes the appropriate literature (references) for the paper topic;

-        The authors contribution is well highlighted in the paper (in Introduction);

-        The method used is correct.

Recommendations:

1)     The authors should better explain the choice of the study (with references).

2) The Conclusion section is too sketchy and the authors must to improve adding comments regarding policy implications of their findings.

3)      I recommend to use professional English editing which would contribute to get a better paper (for example, line 97 “The third paper" - I think that the auhors want to refer to "in the third place, the paper....).

4) Also, that is a different dimension of the used font (sometimes greater - for example, see lines 120-140).

I recommend to use professional English editing which would contribute to get a better paper (for example, line 97 “The third paper" - I think that the auhors want to refer to "in the third place, the paper....).

Author Response

We sincerely thank the reviewers for their careful reading of the manuscript and their recommendations and comments. The paper has been revised in line with all the recommendations, which has made the paper even stronger academically, more coherent and clearer.

The article was also submitted for English proofreading.

The responses and corrections to each reviewer's comments are given in the table below. The corrections have also been highlighted in yellow in the article itself.

Reviewer 2 Report

The article concerns the extremely important problem of the competitiveness of countries in the perspective of 2050. My comments:

1. Is it really possible to talk about such a distant perspective on the basis of short-term data?

2. In Figure 4, each country in each cluster has a different direction. So is it correct to group into such clusters since the behavior is different. In addition, general conclusions are drawn in the summary. Are there real grounds for this?

3. In Model (1) DR is not explained.

4. In my opinion, Fig.2 is very misleading because it does not take into account the size of the country (population, GDP). Meanwhile, the whole idea of the study is built on this.

Author Response

(The authors gave the same response as above.)

Reviewer 3 Report

Dear Authors,

Many thanks for your submission and the opportunity to read your manuscript. There are some ideas that can be useful for your manuscript improvement: 

1. Unclear from where a gap mentioned in the abstract is coming as there is no evidence.  It will be good to add a few sentences on methodology in the abstract;

2. Starting from the first sentence in the introduction it will be good to explain to a broader audience what ambition the EU has regarding 2050. 

3. You have a statement that: Government regulations that mandate pollution control serve as a catalyst 26 for technological and social innovations, fostering the modernisation of industry and infrastructure as well as behaviour changes in society.  But the literature on regulation states that innovations usually come first regulation. 

4. You have competently described the research done but now the new trends in research are coming. For example: on climate change and resilience plans in EU member states. See for enlistment: Kotseva-Tikova, M., & Dvorak, J. (2022). Climate Policy and Plans for Recovery in Bulgaria and Lithuania. Romanian Journal of European Affairs22(2).

5. n the literature review, you presented two concepts: sustainability competitiveness and green competitiveness. However, the distinction between them is not sufficiently well-argued. Perhaps it is better to present a comparison in a table according to certain criteria

5. There are many and various factors influencing green or sustainable competitiveness, and every time of period changes much of the thinking of strategists and policy makers. Unclear statement and what are the factors???

6.  Unclear why do you need it and how it help you in your arguments. 2.2. Place specificity impact to green competitiveness

7. In methodology you did not mention the UK, but it was also a part of the EU for the period under your investigation. 

8. Pls provide which data you used from the following institutions: Data sources were Eurostat, World Bank, and Transparency International

9. It will be good to have policy implications in the conclusions.

All the best

Author Response

(The authors gave the same response as above.)

Reviewer 4 Report

This paper uses environmental performance as an endogenous factor to expand the concept of competitiveness. Through the definition of green competitiveness, the article constructs the corresponding index system. Then the article measures the change in EU countries' green competitiveness. The topic selection has certain theoretical and practical significance. The structure of the article is complete and the terminology is standardized, which reflects the author's expressive ability. However, there are still obvious deficiencies in the following aspects, which still need to be improved.

1. The chart format is not standardized. Charts are longer than the margins and do not fit in Figure 1 and Table 1 etc. The window size of Figure 4 should be resized to a uniform size. The readability of Figure 3 is poor, consider whether to replace it with other graphics.

2. Line 268 "To determine which of the 43 selected factors were most important for each of the formed clusters, a PCA analysis was used". PCA analysis is a basic dimensionality reduction method. The main purpose of this method is to replace more variables with fewer variables, thus reflecting most of the information of the original variables. If it is to determine the importance of factors, please further explain whether it is appropriate to use this method in this article.

3. Although the appendix shows the results of the PCA analysis, the interpretation of the results of the PCA analysis does not seem to be reflected in the article.

4. The green competitiveness improvement strategy is not consistent with the cluster results, resulting in a disconnect between the two.

Author Response

(The authors gave the same response as above.)

Round 2

Reviewer 2 Report

No comments

Author Response

Good morning,

Thank you very much for your comments on how we can improve the quality of this article.

In Figure 1 we deleted extra arrow.

We corrected the font size of line 295 and the line spacing of line 296.

We clarified in Figure 3 the axis.

Sincerely,

Jurgita Bruneckiene

Reviewer 3 Report

Dear Authors,

Thank you for an improved version of your manuscript.

All the best

Author Response

(The authors gave the same response as above.)

Reviewer 4 Report

I'm glad my recommendations can help you. Compared with the previous version, it can be seen that this version has been greatly modified, which makes the logic of the paper more scientific and reasonable. This edition highlights the author's scholarly contributions. I think this version has been greatly improved. But in some details, I think it can be further improved.

1. In Figure 1, there is an extra arrow in the lower left corner;

2. The font size of line 295 is inconsistent, and the line spacing of line 296 is too large;

3. Compared with the previous version, Figure 3 has been greatly improved. But the axis titles and axes still don't seem to correspond. I hope it can be improved.

Author Response

(The authors gave the same response as above.)
